# Feature Reconstruction from Incomplete Tomographic Data without Detour

**Simon Göppel** [1] , **Jürgen Frikel** [2,*] **and Markus Haltmeier** [1]

1 Department of Mathematics, University of Innsbruck, Technikerstraße 13, A-6020 Innsbruck, Austria; simon.goeppel@uibk.ac.at (S.G.); markus.haltmeier@uibk.ac.at (M.H.)
2 Faculty of Mathematics and Computer Sciences, OTH Regensburg, Galgenbergstraße 32, 93053 Regensburg, Germany
* Correspondence: juergen.frikel@oth-regensburg.de

**Abstract:** In this paper, we consider the problem of feature reconstruction from incomplete X-ray CT data. Such incomplete data problems occur when the number of measured X-rays is restricted either due to limit radiation exposure or due to practical constraints, making the detection of certain rays challenging. Since image reconstruction from incomplete data is a severely ill-posed (unstable) problem, the reconstructed images may suffer from characteristic artefacts or missing features, thus significantly complicating subsequent image processing tasks (e.g., edge detection or segmentation). In this paper, we introduce a framework for the robust reconstruction of convolutional image features directly from CT data without the need of computing a reconstructed image first. Within our framework, we use non-linear variational regularization methods that can be adapted to a variety of feature reconstruction tasks and to several limited data situations. The proposed variational regularization method minimizes an energy functional being the sum of a feature dependent data-fitting term and an additional penalty accounting for specific properties of the features. In our numerical experiments, we consider instances of edge reconstructions from angular under-sampled data and show that our approach is able to reliably reconstruct feature maps in this case.

**Keywords:** computed tomography; Radon transform; reconstruction; limited data; sparse data; feature reconstruction; edge detection

## 1. Introduction

Computed tomography (CT) has established itself as one of the standard tools in bio-medical imaging and non-destructive testing. In medical imaging, the relatively high radiation dose that is used to produce high-resolution CT images (and that patients are exposed to) has become a major clinical concern [1–4]. The reduction of the radiation exposure of a patient while ensuring the diagnostic image quality constitutes one of the main challenges in CT. In addition to patient safety, the reduction of scanning times and costs also constitute important aspects of dose reduction, which is often achieved by reducing the X-ray energy level (leading to higher noise levels in the data) or by reducing the number of collected CT data (leading to incomplete data), cf. [1]. Low-dose scanning scenarios are also relevant for in vivo scanning used for biological purposes and for fast tomographic imaging in general. However, due to the limited amount of data, reconstructed images suffer from low signal-to-noise ratio or substantial reconstruction artifacts.

In this work, we particularly consider incomplete data situations, e.g., that arise in a sparse or limited view setup, where CT data is collected only with respect to a small number of X-ray directions or within a small angular range. The intentional reduction of the angular sampling rate leads to an under-determined and severely ill-posed image reconstruction problem, c.f. [5]. As a consequence, the reconstructed image quality can be substantially degraded, e.g., by artefacts or missing features [6], and this can also effect complicate subsequent image processing tasks (such as edge detection or segmentation)

that are often employed within a CAD pipeline (computer aided diagnosis). Therefore, the development of robust feature detection algorithms for CT that ensure the diagnostic image quality is an important and very challenging task. In this paper, we introduce a framework for feature reconstruction directly from incomplete tomographic data, which is in contrast to the classical 2-step approach where reconstruction and feature detection are performed in two separate steps.

### 1.1. Incomplete Tomographic Data

In this article, we consider the parallel beam geometry and use the 2D Radon transform $\mathbf{R}f\colon \mathbb{S}^1 \times \mathbb{R} \to \mathbb{R}$ as a model for the (full) CT data generation process, where $\mathbb{S}^1$ denotes the unit circle in $\mathbb{R}^2$ and $f\colon \mathbb{R}^2 \to \mathbb{R}$ is a function representing the sought tomographic image (CT scan). Here, the value $\mathbf{R}f(\theta, s)$ represents one X-ray measurement over a line in $\mathbb{R}^2$ that is parametrized by the normal vector $\theta \in \mathbb{S}^1$ and the signed distance from the origin $s \in \mathbb{R}$. In what follows, we consider incomplete data situations where the Radon data are available on a circular scanning trajectory and only for a small number of directions, given by $\Theta := \{\theta_1, \ldots, \theta_m\}$. We denote the angularly sampled tomographic Radon data by $\mathbf{R}_\Theta f := (\mathbf{R}f)|_{\Theta \times \mathbb{R}}$. In this context, the (semi-discrete) CT data $\mathbf{R}_\Theta f$ will be called incomplete if the Radon transform is insufficiently sampled with respect to the directional variable. Prominent instances of incomplete data situations are: *sparse angle setup*, where the directions in $\Theta$ are sparsely distributed over the full angular range $[0, \pi]$; annd *limited view setup*, where $\Theta$ covers only small part of the full angular range $[0, \pi]$. Precise mathematical criteria of (in-)sufficient sampling can be derived from the Shannon sampling theory. Those criteria are based on the relation between the number of directions $m = |\Theta|$ and the bandwidth of $f$, cf. [5]. In this work, we will mainly focus on the sparse angle case, with uniformly distributed directions $\theta_1, \ldots, \theta_m$ on a half-circle, e.g., directions $\theta_k := \theta(\varphi_k) = (\cos(\varphi_k), \sin(\varphi_k))^\top$ with uniformly distributed angles $\varphi_k \in [0, \pi)$.

### 1.2. Feature Reconstruction in Tomography

In the following, we consider image features that can be extracted from a CT scan $f \in L^2(\mathbb{R}^2)$ by a convolution with a kernel $U \in L^1(\mathbb{R}^2)$. In this context, the notion of a feature map will refer to the convolution product $f \circledast U$, and the convolution kernel $U$ will be called the feature extraction filter. Examples of feature detection tasks that can be realized by a convolution include edge detection, image restoration, image enhancement, or texture filtering [7]. For example, in the case of edge detection, the filter $U$ can be chosen as a smooth approximation of differential operators, e.g., of the Laplacian operator [8]. In our practical examples, we will mainly focus on edge detection in tomography. However, the proposed framework also applies to more general feature extraction tasks.

In many standard imaging setups, image reconstruction and feature extraction are realized in two separate steps. However, as pointed out in [9], this 2-step approach can lead to unreliable feature maps since feature extraction algorithms have to account for inaccuracies that are present in the reconstruction. This is particularly true for the case of incomplete CT data as those reconstructions may contain artefacts. Hence, combining these two steps into an approach that computes feature maps directly from CT data can lead to a significant performance increase, as was already pointed out in [9,10]. In this work, we account for this fact and extend the results of [9,10] to a more general setting and, in particular, to limited data situations.

### 1.3. Main Contributions and Related Work

In this paper, we propose a framework to directly reconstruct the feature map $U \circledast f$ from the measured tomographic data. Our approach is based on the forward convolution identity for the Radon transform, which is $\mathbf{R}(f \circledast U) = (\mathbf{R}f) \circledast_s (\mathbf{R}U)$, where on the right hand side the convolution is taken with respect to the second variable of the Radon transform, cf. [5]. This identity implies that, given (semi-discrete) CT data, the feature map satisfies the (discretized) equation $\mathbf{R}_\Theta h = y_\Theta$, where $y_\Theta = \mathbf{R}_\Theta f \circledast_s \mathbf{R}_\Theta U$ is the modified

(preprocessed) CT data. Therefore, the sought feature map can be formally computed by applying a discretized version of the inverse Radon transform to $y_\Theta$, i.e., as $h_\Theta = \mathbf{R}_\Theta^{-1}(y_\Theta)$. In the case of full data (sufficient sampling), this can be accurately and efficiently computed by using the well-known filtered backprojection (FBP) algorithm with the filter $\mathbf{R}_\Theta U$. However, if the CT data are incomplete, this approach would lead to unreliable feature maps since in such situations the FBP is known to produce inaccurate reconstruction results, cf. [5,6].

In order to account for data incompleteness, we propose to replace the inverse $\mathbf{R}_\Theta^{-1}$ by a suitable regularization method that is also able to deal with undersampled data. More concretely, we propose to reconstruct the (discrete) feature map $h_\Theta$ by the minimizing the following Tikhonov-type functional:

$$h_\Theta \in \arg\min_h \frac{1}{2} \|\mathbf{R}_\Theta h - u_\Theta \circledast_s y_\Theta\|^2 + r(h)\,.$$

This framework offers a flexible way to incorporate a priori information about the feature map into the reconstruction and, in this way, to account for the missing data. For example, from the theory of compressed sensing, it is well known that sparsity can help to overcome the classical Nyquist–Shannon–Whittaker–Kotelnikov paradigm [11]. Hence, whenever the sought feature map is known to be sparse (e.g., in case of edge detection), sparse regularization techniques can be easily incorporated into this framework.

Approaches that combine image reconstruction and edge detection have been proposed for the case of full tomographic data, e.g., in [9,10]. Although the presented work follows the spirit of [9,10], it comes with several novelties and advantages. On a formal level, our approach is based on the forward convolution identity, in contrast to the dual convolution identity, given by $(\mathbf{R}^* u) \circledast f = \mathbf{R}^*(u \circledast_s \mathbf{R} f)$, that is employed in [9,10]. The latter requires full (properly sampled) data, since the backprojection operator $\mathbf{R}^*$ integrates over the full angular range (requiring proper sampling in the angular variable). In contrast, our framework is applicable to incomplete Radon data situations, since the forward convolutional identity (used in our approach) can be applied to more general situations. Moreover, in order to recover the feature map $U \circledast f$, we use non-linear regularization methods that can be adapted to a variety of situations and incorporate different kinds of prior information. From this perspective, our approach also offers more flexibility. A similar approach was presented in our recent proceedings article [12], where the main focus was on the stable recovery of the image gradient from CT data and its application to Canny edge detection. Following the ideas of [9,10], similar feature detection methods were also developed for other types of tomography problems, e.g., in [13–15]. Besides that, we are not aware of any further results concerning convolutional feature reconstruction from incomplete X-ray CT data.

Combinations of reconstruction and segmentation have also been presented in the literature for different types of tomography problems, e.g., in [16–22]. As a commonality to our approach, many of those methods are based on the minimization of an energy functional of the form $\|\mathbf{R}_\Theta f - y\|^2 + r(f * \mathbf{U})$, incorporating feature maps as prior information. Important examples include Mumford–Shah-like approaches [17,19,21,22] or the Potts model [18]. Additionally, geometric approaches for computing segmentation masks directly from tomographic data were employed in [16].

### 1.4. Outline

Following the introduction in Section 1, Section 2 provides some basic facts about the Radon transform, sampling and sparse recovery. In Section 3, we introduce the proposed feature reconstruction framework and present several examples of convolutional feature reconstruction filters, along with corresponding data filters, mainly focusing on the case of edge detection. Experimental results will be presented in Section 4. We conclude with a summary and outlook given in Section 5.

## 2. Materials and Methods

In this section, we recall some basic facts about the 2D Radon transform, including important identities and sampling conditions. In particular, we define the sub-sampled Radon transform that will be used throughout this article. Although, our presentation is restricted to the 2D case (because this makes the presentations more concise and clear), the presented concepts can be easily generalized to the $d$-dimensional setup.

### 2.1. The Radon Transform

Let $\mathcal{S}(\mathbb{R}^2)$ denote the Schwartz space on $\mathbb{R}^2$ (space of smooth functions that are rapidly decaying together with all their derivatives) and $\mathcal{S}(\mathbb{S}^1 \times \mathbb{R})$ denote the Schwartz space over $\mathbb{S}^1 \times \mathbb{R}$ as the space of all smooth functions that are rapidly decaying together with all their derivatives in the second component, cf. [5]. We consider the Radon transform as an operator between those Schwartz spaces, $\mathbf{R} \colon \mathcal{S}(\mathbb{R}^2) \to \mathcal{S}(\mathbb{S}^1 \times \mathbb{R})$, which is defined via

$$\mathbf{R}f(\theta, s) := \int_{-\infty}^{\infty} f(s\theta + t\theta^{\perp})\mathrm{d}t, \tag{1}$$

where $s \in \mathbb{R}$, $\theta \in \mathbb{S}^1$ and $\theta^{\perp}$ denotes the rotated version of $\theta$ by $\pi/2$ counterclockwise (in particular, $\theta^{\perp}$ is a unit vector perpendicular to $\theta$). The value $\mathbf{R}f(\theta, s)$ represents one X-ray measurement along the X-ray path that is given by the line $L(\theta, s) = \{x \in \mathbb{R}^2 : \langle x, \theta \rangle = s\}$. Since $L(-\theta, -s) = L(\theta, s)$, the following symmetry property holds for the Radon transform, $\mathbf{R}f(-\phi, -s) = \mathbf{R}f(\theta, s)$. Hence, it is sufficient to know the values of Radon transform only on a half-circle. Such data is therefore considered to be complete. The dual transform (backprojection operator) is defined as $\mathbf{R}^* : \mathcal{S}(\mathbb{S}^1 \times \mathbb{R}) \to \mathcal{S}(\mathbb{R}^2)$,

$$\mathbf{R}^* g(x) := \int_{\mathbb{S}^1} g(\theta, \theta \cdot x)\mathrm{d}\theta. \tag{2}$$

The Radon transform is a well defined linear and injective operator, and several analytic properties are well-known. One of the most important properties is the so-called Fourier slice theorem that describes the relation between the Radon and the Fourier transforms. In order to state this relation, we first recall that the Fourier transform is defined as $\mathbf{F} : \mathcal{S}(\mathbb{R}^d) \to \mathcal{S}(\mathbb{R}^d)$, $\mathbf{F}f(\xi) := (2\pi)^{-d/2} \int_{\mathbb{R}^d} f(x)e^{-ix\cdot\xi}\,\mathrm{d}x$ for $d \in \mathbb{N}$. Whenever convenient, we will also use the abbreviated notation $\hat{f}(\xi) := \mathbf{F}f(\xi)$. The Fourier transform is a linear isomorphism on the Schwartz space $\mathcal{S}(\mathbb{R}^d)$, and its inverse is given by $\check{f}(x) := \mathbf{F}^{-1}f(x) = (2\pi)^{-d/2} \int_{\mathbb{R}^2} f(\xi)e^{ix\cdot\xi}\,\mathrm{d}\xi$. In what follows, we will denote the convolution of two functions $f, g : \mathbb{R}^d \to \mathbb{R}$ by $f \circledast g(x) := \int_{\mathbb{R}^d} f(x - y)g(y)\mathrm{d}y$, where $d \in \mathbb{N}$. Moreover, for functions $g \in \mathcal{S}(\mathbb{S}^1 \times \mathbb{R})$, the Fourier transform $\mathbf{F}_s g$ will refer to the 1D-Fourier transform of $g$ with respect to the second variable. Analogously, $g \circledast_s h$ will denote the convolution of $g, h : \mathbb{S}^1 \times \mathbb{R} \to \mathbb{R}$ with respect to the second variable.

**Lemma 1** (Properties of the Radon transform).

(R1) *Fourier slice theorem:* $\forall f \in \mathcal{S}(\mathbb{R}^2) \; \forall (\theta, s) \in \mathbb{S}^1 \times \mathbb{R} \colon \mathbf{F}_s \mathbf{R}f(\theta, \sigma) = \sqrt{2\pi} \cdot \mathbf{F}f(\theta\sigma)$.

(R2) *Convolution identity:* $\forall U, f \in \mathcal{S}(\mathbb{R}^2) \colon \mathbf{R}(f \circledast U) = \mathbf{R}f \circledast_s \mathbf{R}U$.

(R3) *Dual convolution identity:* $\forall u \in \mathcal{S}(\mathbb{S}^1 \times \mathbb{R}) \; \forall f \in \mathcal{S}(\mathbb{R}^2) \colon \mathbf{R}^*u \circledast f = \mathbf{R}^*(u \circledast_s \mathbf{R}f)$.

(R4) *Intertwining with Derivatives:* $\forall \alpha \in \mathbb{N}^2 \; \forall f \in \mathcal{S}(\mathbb{R}^2) \colon \mathbf{R}\partial_x^{\alpha} f = \theta^{\alpha}\partial_s^{|\alpha|}\mathbf{R}f$

(R5) *Intertwining with Laplacian:* $\forall f \in \mathcal{S}(\mathbb{R}^2) \colon \mathbf{R}\Delta_x f = \partial_s^2 \mathbf{R}f$.

**Proof.** All identities are derived in [5] (Chapter II). □

The approach that we are going to present in Section 3 is based on the convolution identity (R2) and can be formulated for an arbitrary spatial dimension $d \geq 2$. For the sake of clarity we consider two spatial dimensions $d = 2$. In this case, we will use the parametrization of $\mathbb{S}^1$ given by $\theta(\varphi) := (\cos(\varphi), \sin(\varphi))^{\top}$ with $\varphi \in [0, \pi)$. Then

$\theta^\perp(\varphi) = (-\sin(\varphi), \cos(\varphi))^\top$. For the Radon transform, we will (with some abuse of notation) write

$$\mathbf{R}f(\varphi, s) := \mathbf{R}f(\theta(\varphi), s).$$

### 2.2. Sampling the Radon Transform

Since in practice one has to deal with discrete data, we are forced to work with discretized (sampled) versions of the Radon transform. In this context, questions about proper sampling arise. In order to understand what it means for the CT data to be complete (properly sampled) or incomplete (improperly sampled), we recall some basic facts from the Shannon sampling theory for the Radon transform for the case of parallel scanning geometry (see for example [5] (Section III)).

In what follows, we assume that $f$ is compactly supported on the unit disc $D \subseteq \mathbb{R}^2$ and consider sampled CT data $\mathbf{R}f(\varphi_j, s_l)$ with $N_\varphi \in \mathbb{N}$ equispaced angles $\varphi_j$ in $[0, \pi)$ and $N_s$ equispaced values $s_l$ in $[-1, 1]$ for the $s$-variable, i.e.,

$$(\varphi_j, s_\ell) = \left( \frac{j\pi}{N_\varphi}, \frac{\ell}{N_s} \right) \quad \text{for } (j, \ell) \in \{0, \dots, N_\varphi - 1\} \times \{-N_s, \dots, N_s\}. \tag{3}$$

For the given sampling points (3) and a finite dimensional subspace $\mathbb{X}_0 \subseteq \mathcal{S}(\mathbb{R}^d)$, we define the *discrete Radon transform* as

$$\mathrm{R} \colon \mathbb{X}_0 \to \mathbb{R}^{N_\varphi \times (2N_s + 1)} \colon f \mapsto (\mathbf{R}f(\theta_j, s_\ell))_{j,\ell}. \tag{4}$$

The basic question of classical sampling theory in the context of CT is to find conditions on the class of images $f \in \mathbb{X}_0$ and on the sampling points under which the sampled data $\mathrm{R}f$ uniquely determines the unknown function $f$. Sampling theory for CT has been studied, for example, in [23–27]. While the classical sampling theory (e.g., in the setting of classical signal processing) works with the class of band-limited functions, the sampling conditions in the context of CT are typically derived for the class of essentially band-limited functions.

**Remark 1** (Band-limited and essentially band-limited functions). *A function $f \in L^2(\mathbb{R}^2)$ is called b-band-limited if its Fourier transform $\mathbf{F}f(\xi)$ vanishes for $\|\xi\| > b$. A function $f$ is called essentially b-band-limited if $\hat{f}(\xi)$ is negligible for $\|\xi\| \geq b$ in the sense that $\epsilon_0(f, b) := \int_{\|\xi\| \geq b} |\mathbf{F}f(\xi)| d\xi$ is sufficiently small; see [5]. One reason for working with essentially band-limited functions in CT is that functions with compact support cannot be strictly band-limited. However, the quantity $\epsilon_0(f, b)$ can become arbitrarily small for functions with compact support.*

The bandwidth $b$ is crucial for the correct sampling conditions and the calculation of appropriate filters. If $\mathbb{X}_0$ consists of essentially $b$-band-limited functions that vanish outside the unit disc $D$, then the correct sampling conditions are given by [5]

$$(N_\varphi, N_s) := \left( \lceil b \rceil, \lceil b/\pi \rceil \right). \tag{5}$$

Obviously, as the bandwidth $b$ increases, the step sizes $\pi/N_\varphi$ and $1/N_s$ have to decrease in order such that (5) is satisfied. Thus, if the bandwidth $b$ is large, a large number measurements (roughly $2b^2/\pi$) have to be collected. As a consequence, for high-resolution imaging, the sampling conditions require a large number of measurements. Thus, in practical applications, high-resolution imaging in CT also leads to large scanning times and to high doses of X-ray exposure. A classical approach for dose reduction consists of the reduction of X-ray measurements. For example, this can be achieved by angular undersampling, where Radon data is collected only for a relatively small number of directions $\Theta \subseteq \{\theta_0, \dots, \theta_{N_\varphi - 1}\}$.

**Definition 1** (Sub-sampled Radon transform). *Let $(N_\varphi, N_s)$ be defied by (5) and let $\mathbb{X}_0$ be the set of essentially b-band-limited functions that vanishes outside the unit disc $D$ (note that in that*

*case, the discrete Radon transform defined in* (4) *is correctly sampled). For $\Theta \subseteq \{\theta_0, \ldots, \theta_{N_\varphi - 1}\}$, we call*

$$\mathrm{R}_\Theta \colon \mathbb{X}_0 \to \mathbb{R}^{|\Theta| \times (2N_s + 1)} \colon f \mapsto (\mathrm{R}f)|_{\Theta \times \{-N_s, \ldots, N_s\}} \tag{6}$$

*the sub-sampled discrete Radon transform. We will also use the semi-discrete Radon transform $\mathbf{R}_\Theta f := (\mathbf{R}f)|_{\Theta \times \mathbb{R}}$, where we only sample in the angular direction but not in the radial direction.*

If we perform actual undersampling, where the number of directions in $\Theta$ is much less than $N_\varphi$, then the linear equation $\mathrm{R}_\Theta f = y_\Theta$ will be severely under-determined, and its solution requires additional prior information (e.g., sparsity of the feature map).

## 3. Feature Reconstruction from Incomplete Data

In this section, we present our approach for feature map reconstruction from incomplete data. For a given bandwidth $b$, we let $\mathbb{X}_0$ denote the set of essentially $b$-band-limited functions that vanishes outside $D$. Furthermore, we assume that the set of directions $\{\theta_0, \ldots, \theta_{N_\varphi - 1}\}$ is chosen according to the sampling conditions (5).

**Problem 1** (Feature reconstruction task). *Let $\Theta \subseteq \{\theta_0, \ldots, \theta_{N_\varphi - 1}\}$ and let $y_\Theta \colon \Theta \times \mathbb{R} \to \mathbb{R}$ be the noisy subsampled (semi-discrete) CT data with $\|\mathbf{R}_\Theta f - y_\Theta\| \leq \delta$, where $f \in \mathbb{X}_0$ is the true but unknown image and $\delta > 0$ is the known noise level. Given a feature extraction filter $U \colon \mathbb{R}^2 \to \mathbb{R}$, our goal is to estimate the feature map $U \circledast f$ from the (undersampled) data $y_\Theta$.*

**Remark 2.**
1. *From a general perspective, Problem 1 is related to the field of optimal recovery [28], where the goal is to estimate certain features of an element in a space $\mathbb{X}_0$ from noisy indirect observations;*
2. *Depending on the particular choice of the filter $U$, Problem 1 corresponds to several typical tasks in tomography. For example, if $U$ is chosen as an approximation of the Delta distribution, Problem 1 is equivalent to the classical image reconstruction problem. In fact, the filtered backprojection algorithm (FBP) is derived in this way from the dual convolution identity (R3) for the full data case, cf. [5]. Another instance of Problem 1 is edge reconstruction from tomographic data $y_\Theta$. For example, this can be achieved by choosing the feature extraction filter $U$ as the Laplacian of an approximation to the Delta distribution (e.g., Laplacian of Gaussian (LoG)). Then, Problem 1 boils down to an approximate recovery of the Laplacian of $f$, which is used in practical edge-detection algorithms (e.g., LoG-filter [7,8]);*
3. *Traditionally, the solution of Problem 1 is realized via the 2-step approach: First, by estimating $f$ and, secondly, by applying convolution in order to estimate the feature map $U \circledast f$. This 2-step approach has several disadvantages: Since image reconstruction in CT is (possibly severely) ill-posed, the fist step might introduce huge errors in the reconstructed image. Those errors will also be propagated through the second (feature extraction) step, which itself can be ill-posed and even further amplify errors. In order to reduce the error propagation of the first step, regularization strategies are usually applied. The choice of a suitable regularization strategy strongly depends on the particular situation and on the available prior information about the sought object $f$. However, the recovery of $f$ requires different prior knowledge than feature extraction. This mismatch can lead to a substantial loss of performance in the feature detection step;*
4. *In order to overcome the limitations mentioned in the remark above, image reconstruction and edge detection were combined in [9,10], where explicit formulas for estimating the edge map have been derived using the method of approximate inverse. This approach is also based on the dual convolution identity (R3) and is closely related to the standard filtered backprojection (FBP) algorithm. However, this approach is not applicable to the case of undersampled data, since [9,10] employ the dual convolutional identity (R3) and calculate the reconstruction filters of the form $\mathbf{R}_\Theta^* u$. In this calculation, in order to achieve a good approximation of the integral in (2), a properly sampled Radon data is required.*

To overcome the limitations mentioned in the remark above, we derive a novel framework for feature reconstruction in the next subsection (based on the forward convolutional identity (R3)) that does not make use of the continuous backprojection and, hence, can be applied to more general situations.

### 3.1. Proposed Feature Reconstruction

Our proposed framework for solving the feature reconstruction Problem 1 is based on the forward convolution identity (R2) stated in Lemma 1. Because the convolution on the right-hand side of (R2) acts only on the second variable, the convolution identity is not affected by the subsampling in the angular direction. Therefore, we have

$$\mathbf{R}_\Theta(f \circledast U) = u_\Theta \circledast_s \mathbf{R}_\Theta f \quad \text{with } u_\Theta := \mathbf{R}_\Theta U. \tag{7}$$

Formally, the solution of (7) takes the form $f \circledast U = \mathbf{R}_\Theta^{-1}(u_\Theta \circledast_s \mathbf{R}_\Theta f)$. If the data are properly sampled, this can be accurately and efficiently computed by applying the FBP algorithm to the filtered CT data $y_\Theta = u_\Theta \circledast_s \mathbf{R}_\Theta f$. In this context, the data filter $u_\Theta$ needs to be precomputed (from a given feature extraction filter $U$) in a filter design step. However, if the data $\mathbf{R}_\Theta f$ are not properly sampled, the Equations (7) are underdetermined and, in this case, FBP does not produce accurate results, cf. [5,6]. In order to account for data incompleteness and to stably approximate the feature map $f \circledast U$, a priori information about the specific feature kernel $U$ or the feature map $f \circledast U$ needs to be integrated into the reconstruction procedure. As a flexible way for doing this, we propose to approximate the inverse $\mathbf{R}_\Theta^{-1}$ by the following variational regularization scheme:

$$\frac{1}{2}\|\mathbf{R}_\Theta h - u_\Theta \circledast_s y_\Theta\|_2^2 + r(h) \to \min_{h \in \mathbb{X}_0}. \tag{8}$$

Here, $y_\Theta : \Theta \times \mathbb{R} \to \mathbb{R}$ denotes the noisy (semi-discrete data), and $r : \mathbb{X}_0 \to [0, \infty]$ is a regularization (penalty) term.

**Example 1.**

1.  IMAGE RECONSTRUCTION: *Here, the feature extraction filter $U = U_\alpha$ is chosen as an approximation to the Delta distribution. For example, as $U = g_\alpha$ with*

    $$g_\alpha(x) = \frac{1}{2\pi\alpha^2} \exp\left(-\frac{\|x\|^2}{2\alpha^2}\right), \quad \alpha > 0 \tag{9}$$

    *being the Gaussian kernel. Another way of choosing $U$ for reconstruction purposes is through ideal low-pass filters $U_\alpha$ that are defined in the frequency domain via $\mathbf{F}U_\alpha = \chi_{D(0,\alpha^{-1})}$, where $\alpha > 0$, $D(0, \alpha^{-1}) \subset \mathbb{R}^2$ denotes a ball in $\mathbb{R}^2$ with radius $1/\alpha$, and $\chi_A$ is the characteristic function of the set $A \subseteq \mathbb{R}^2$. It can be shown that in both cases, $U_\alpha \circledast_s f \to f$ as $\alpha \to 0$. These filters and its variants are often used in the context of the FBP algorithm.*

2.  GRADIENT RECONSTRUCTION: *Here $U = U_\alpha$ is chosen as a partial derivative of an approximation of the Delta distribution. For example, as $U_\alpha = (U_\alpha^{(1)}, U_\alpha^{(2)})$ with $U_\alpha^{(i)} := \frac{\partial g_\alpha}{\partial x_i}$, $i = 1, 2$. This way, one obtains an approximation of the gradient of $f$ via*

    $$\nabla_x f = (U_\alpha^{(1)} \circledast f, U_\alpha^{(2)} \circledast f) =: U_\alpha \circledast f,$$

    *where in the last equation above we applied the convolution $\circledast$ componentwise. Such approximations of the gradient are, for example, used inside the well-known Canny edge detection algorithm [29].*

3.  LAPLACIAN RECONSTRUCTION: *Analogously to the gradient approximation, $U$ is chosen to be the Laplacian of an approximation to the Delta distribution. A prominent example, is*

*the Laplacian of Gaussian (LoG), i.e., $U_\alpha = \Delta_x g_\alpha$, also known as the Marr–Hildreth operator. This operator is also used for edge detection, corner detection and blob detection, cf. [30].*

Depending on the problem at hand, there are several different ways of choosing the regularizer $r(h)$. Prominent examples in the case of image reconstruction include total variation (TV) or the $\ell^1$ norm (possibly in the context of some basis of frame expansion). For the reconstruction of the derivatives (or edges in general), we will use the $\ell^1$ norm as the regularization term because derivatives of images can be assumed to be sparse and because the problem (8) can be efficiently solved in this case.

*3.2. Filter Design*

The first step in our framework is a filter design for (8). That is, given a feature extraction kernel $U$, we first need to calculate the corresponding filter $u_\Theta = \mathbf{R}_\Theta U$ for the CT data, cf. (7). In our setting, filter design therefore amounts to the evaluation of the Radon transform of $U$. In contrast to our approach, the filter design step of [9] consists of calculating a solution of the dual equation $U = \mathbf{R}^* u$ given the feature extraction filter $U$. As discussed above, the latter case requires full data and might be computationally more involved. From this perspective, filter design required by our approach offers more flexibility and can be considered somewhat simpler.

We now discuss some of the Examples 1 in more detail and calculate the associated CT data filters $u_\Theta$. In particular, we focus on the Gaussian approximations of the Delta distributions stated in (9). In a first step, we compute the Radon transform of a Gaussian.

**Lemma 2.** *The Radon transform of the Gaussian $g_\alpha$, defined by (9), is given by*

$$\mathbf{R} g_\alpha(\varphi, s) = \frac{1}{\alpha \sqrt{2\pi}} \cdot \exp\left(-\frac{s^2}{2\alpha^2}\right). \tag{10}$$

Since the Gaussian $g_\alpha$ converges to the Delta distribution as $\alpha \to 0$, the smoothed version $f_\alpha := f \circledast g_\alpha$ constitutes an approximation to $f$ for small values of $\alpha$. In order to obtain approximations to partial derivatives of $f$, we note that $\frac{\partial f_\alpha}{\partial x_i} = f \circledast \frac{\partial g_\alpha}{\partial x_i}$. Hence, using the feature extraction filters $U_\alpha^{(i)} := \frac{\partial g_\alpha}{\partial x_j}$, the Problem 1 amounts to reconstructing partial derivatives of $f$. Using this observation together with Lemma 2 and the property (R4), we can explicitly calculate data filters used in different edge reconstruction algorithms (such as Canny or for the Marr–Hildreth operator).

**Proposition 1.** *Let the Gaussian $g_\alpha$ be defined by (9).*

1.  GRADIENT RECONSTRUCTION: *For the feature extraction filter $U_{\text{grad}} := \nabla_x g_\alpha$, the corresponding data filter $u_{\text{grad}} = (u_\alpha^{(1)}, u_\alpha^{(2)})$ is given by*

$$u_{\text{grad}}(\varphi, s) = \mathbf{R} U_{\text{grad}}(\varphi, s) = -\frac{s}{\alpha^3 \sqrt{2\pi}} \cdot \exp\left(-\frac{s^2}{2\alpha^2}\right) \cdot \theta(\varphi) \tag{11}$$

*Note that in (11), the notation $\mathbf{R} U_{\text{grad}}$ refers to a vector-valued function that is defined by a componentwise application of the Radon transform (cf. Example 1, No. 2).*

2.  LAPLACIAN RECONSTRUCTION: *For the feature extraction filter $U_\alpha := \Delta_x g_\alpha$, the corresponding data filter is given by*

$$u_{\text{LoG}}(\varphi, s) = \mathbf{R} U_{\text{LoG}}(\varphi, s) = \frac{1}{\alpha^3 \sqrt{2\pi}} \cdot \exp\left(-\frac{s^2}{2\alpha^2}\right) \cdot \left(\frac{s^2}{\alpha^2} - 1\right) \tag{12}$$

From Proposition 1, we immediately obtain an explicit reconstruction formula for the approximate computation of the gradient and of the Laplacian of $f \in \mathcal{S}(\mathbb{R}^2)$:

$$\nabla_x f_\alpha = \mathbf{R}^{-1}(u_{\text{grad}} \circledast_s \mathbf{R} f) \quad \text{and} \quad \Delta_x f_\alpha = \mathbf{R}^{-1}(u_{\text{LoG}} \circledast_s \mathbf{R} f).$$

Both of the above formulas are of the FBP type and can be implemented using the standard implementations of the FBP algorithm with a modified filter. This approach has the advantage that only one data-filtering step has to be performed, followed by the standard backprojection operation.

In order to derive FBP filters for the gradient and Laplacian reconstruction, let us first note that $\mathbf{R}^{-1} = \mathbf{R}^* \circ \Lambda$, where the operator $\Lambda$ acts on the second variable and is defined in the Fourier domain by $\mathbf{F}_s(\Lambda g)(\varphi, \omega) = (4\pi)^{-1} \cdot |\omega| \cdot (\mathbf{F}_s g)(\varphi, \omega)$ for $g \in \mathcal{S}(\mathbb{S}^1 \times \mathbb{R})$, cf. [5]. Now, using the relations for the Fourier transform in 1D, $\mathbf{F}(\mathrm{d}f/\mathrm{d}x)(\omega) = i \cdot \omega \cdot \hat{f}(\omega)$, $\mathbf{F}(\mathrm{d}^2 f/\mathrm{d}x^2)(\omega) = -\omega^2 \hat{f}(\omega)$ and $\mathbf{F}(f * g) = \sqrt{2\pi} \cdot \hat{f} \cdot \hat{g}$. Together with

$$\mathbf{F}_s(\mathbf{R} g_\alpha)(\varphi, s) = \frac{1}{\sqrt{2\pi}} \cdot \exp\left(-\frac{\alpha^2 s^2}{2}\right),$$ (13)

we obtain the following result.

**Proposition 2.** *Let the FBP filters $W_{\text{grad}} = W_{\text{grad}}(\varphi, s)$ and $W_{\text{LoG}} = W_{\text{LoG}}(\varphi, s)$ be given in the Fourier domain (componentwise) by*

$$\mathbf{F}_s W_{\text{grad}}(\varphi, \omega) = \frac{1}{4\pi} \cdot i \cdot \omega \cdot |\omega| \cdot \exp\left(-\frac{-\alpha^2 s^2}{2}\right) \cdot \theta(\varphi),$$ (14)

*and*

$$\mathbf{F}_s W_{\text{grad}}(\varphi, \omega) = -\frac{1}{4\pi} \cdot |\omega|^3 \cdot \exp\left(-\frac{-\alpha^2 s^2}{2}\right),$$ (15)

*where $\varphi \in (0, 2\pi)$ and $\omega \in \mathbb{R}$. Then, for $f \in \mathcal{S}(\mathbb{R}^2)$, we have*

$$\nabla_x f_\alpha = \mathbf{R}^*(W_{\text{grad}} \circledast_s \mathbf{R} f) \quad \text{and} \quad \Delta_x f_\alpha = \mathbf{R}^*(W_{\text{LoG}} \circledast_s \mathbf{R} f).$$ (16)

Since the FBP algorithm is a regularized implementation of $\mathbf{R}^{-1}$ (cf. [5]), a standard toolbox implementation could be used in practice in order to compute $\nabla_x f_\alpha$ and $\Delta_x f_\alpha$. To this end, one only needs to use the modified filters for the FBP, provided in (14) and (15), instead of the standard FBP filter (such as Ram-Lak). Again, let us emphasize that the reconstruction formulae (16) can only be used in the case of properly sampled CT data. If the CT data does not satisfy the sampling requirements, e.g., in case of angular undersampling, this FBP algorithm will produce artifacts which can substantially degrade the performance of edge detection. In such cases, our framework (8) should be used in combination with a suitable regularization term. In the context of edge reconstruction, we propose to use $\ell^1$ regularization in combination with $\ell^2$ regularization. This approach will be discussed in the next section.

So far, we have constructed data filters for the approximation of the gradient and Laplacian in the spatial domain, cf. Proposition 1, and derived according to FBP filters in the Fourier domain in Proposition 2. In a similar fashion, one can derive various related examples by replacing the Gaussian by feature kernels whose Radon transform is known analytically. Another way of obtaining practically relevant data filters (for a wide class of feature filters) is to derive expressions for the data filters in the Fourier domain (i.e., filter design in the Fourier domain). In the following, we provide two basic examples for filter design in the Fourier domain. To this end, we will employ the Fourier slice theorem, cf. Lemma 1, (R1).

**Remark 3.**

1. LOWPASS LAPLACIAN: *The Laplacian of the ideal lowpass is defined as*

$$U_b = \Delta_x \mathbf{F}^{-1}(\chi_{D(0,b)}),$$

*where b is the bandwidth of $U_b$. Using the property (R5), we get $\mathbf{R}(U_b) = \frac{\partial^2}{\partial s^2} \mathbf{R}(\mathbf{F}^{-1}(\chi_{D(0,b)}))$. By the Fourier slice theorem, we obtain*

$$\mathbf{F}_s(\mathbf{R}(U_b))(\varphi, \omega) = -\omega^2 \chi_{D(0,b)}(\omega \cdot \theta(\varphi)) = -\omega^2 \chi_{[-b,b]}(\omega).$$

*Hence, the associated data filter is given by*

$$u_b(\varphi, s) := \mathbf{R}U_b(\varphi, s) = \frac{\partial^2}{\partial s^2} \mathbf{F}_s^{-1}(\chi_{[-b,b]})(s) = \sqrt{\frac{2}{\pi}} \cdot \frac{\partial^2}{\partial s^2} \frac{\sin(bs)}{s}$$

$$= \sqrt{\frac{2}{\pi}} \cdot \left( \frac{2\sin(bs)}{s^3} - \frac{2b\cos(bs)}{s^2} - \frac{b^2\sin(bs)}{s} \right). \tag{17}$$

*Because $u_b$ is b-band-limited, the convolution with the filter (17) can be discretized systematically whenever the underlying image is essentially b-band-limited. To this end, assume that the function f has bandwidth b. Then, $y = \mathbf{R}f$ has bandwidth b as well (with respect to the second variable), and therefore, the continuous convolution $\mathbf{R}f \circledast_s u_b$ can be exactly computed via discrete convolution. Using discretization (3) and taking $s_\ell = \frac{\pi}{b} \cdot \ell$, we obtain from (17) the discrete filter*

$$u_b(\varphi, s_\ell) = -\sqrt{\frac{2}{\pi}} \cdot b^3 \cdot \begin{cases} \dfrac{1}{3}, & \text{if} \quad \ell = 0, \\[2mm] \dfrac{2 \cdot (-1)^\ell}{\pi^2 \ell^2} & \text{if} \quad \ell \neq 0. \end{cases} \tag{18}$$

*According to one-dimensional Shannon sampling theory, we compute $y \circledast_s u_b$ via discrete convolution with the filter coefficients given in (18).*

2. RAM–LAK-TYPE FILTER: *Consider the feature extraction filter*

$$U_{b,1} = \Delta_x \mathbf{F}^{-1}\left[ \chi_{D(0,b)} \cdot (1 - \| \cdot \|)_+ \right],$$

*where $(1 - \| \cdot \|)_+ := \max\{0, 1 - \| \cdot \|\}$. Note that for $b \geq 1$, we have $u_{b,1} = u_{1,1}$, since in this case $\chi_{D(0,b)} \cdot (1 - \| \cdot \|)_+ = (1 - \| \cdot \|)_+$. Hence, we consider the case $b \leq 1$. In a similar fashion as above, we obtain*

$$u_{b,1} := \mathbf{R}U_{b,1}(\varphi, s) = \frac{\partial^2}{\partial s^2} \mathbf{F}_s^{-1}\left[ \chi_{[-b,b]} \cdot (1 - |\cdot|) \right](s)$$

$$= \frac{\partial^2}{\partial s^2} \left[ \mathbf{F}_s^{-1}[\chi_{[-b,b]}](s) - \mathbf{F}_s^{-1}[|\cdot| \cdot \chi_{[-b,b]}](s) \right]. \tag{19}$$

*Evaluating $u_{b,1}$ at $s_\ell = \frac{\pi}{b} \cdot \ell$, we get*

$$u_{b,1}(\theta, s_\ell) = \sqrt{\frac{2}{\pi}} \cdot b^3 \cdot \begin{cases} \dfrac{3b - 4}{12} & \text{if } \ell = 0 \\[2mm] \dfrac{3b - 2}{\pi^2 \ell^2} & \text{if } \ell \text{ is even} \\[2mm] -\dfrac{3b - 2}{\pi^2 \ell^2} + \dfrac{12b}{\pi^4 \ell^4} & \text{if } \ell \text{ is odd .} \end{cases} \tag{20}$$

*Again, we can evaluate $y \circledast_s u_{b,1}$ via discrete convolution with the filter coefficients (20).*

Finally, let us note that there are several other examples for feature reconstruction filters for which one can derive explicit formulae of corresponding data filters in a similar way as we did in this section, for example, in the case of approximation of Gaussian derivatives of higher order or for band-limited versions of derivatives.

## 4. Numerical Results

In our numerical experiments, we focus on the reconstruction of edge maps. To this end, we use our framework (8) in combination with feature extraction filters that we have derived in Proposition 1 and in Remark 3. Since the gradient and the Laplacian of an image have relatively large values only around edges and small values elsewhere, we aim at exploiting this sparsity and, hence, use a linear combination $r(h) = \mu \|\nabla h\|_2^2 + \lambda \|h\|_1$ as a regularizer in (8). The resulting minimization problem then reads

$$\frac{1}{2} \|\mathbf{R}_\Theta h - u_\Theta \circledast_s y_\Theta\|_2^2 + \mu \|\nabla h\|_2^2 + \lambda \|h\|_1 \to \min_{h \in \mathbb{X}_0} . \tag{21}$$

If $\mu = 0$, this approach reduces to the $\ell^1$ regularization which is known to favor a sparse solution. If $\mu \neq 0$, the additional $H^1$-term increases the smoothness of the recovered edges. In order to numerically minimize (21), we use the fast iterative shrinkage-thresholding algorithm (FISTA) of [31]. Here, we apply the forward step to $\frac{1}{2}\|\mathbf{R}_\Theta h - u_\Theta \circledast_s y_\Theta\|_2^2 + \mu \|\nabla h\|_2^2$ and the backward step to $\lambda \|h\|_1$. The discrete $\ell^p$ norms are defined by $\|h\|_p = (\sum_{i,j=1}^N |h_{ij}|^p)^{\frac{1}{p}}$ and the discrete Radon transform $\mathbf{R}_\Theta$ is computed via the composite trapezoidal rule and bilinear interpolation. The adjoint Radon transform $\mathbf{R}_\Theta^*$ is implemented as a discrete backprojection following [5].

### 4.1. Reconstruction of the Laplacian Feature Map

We first investigate the feasibility of the proposed approach for recovering the Laplacian of the initial image. For our first experiment, we use a simple phantom image which is defined as a characteristic function of the union of three (overlapping) discs. For these synthetic data, we obtain precise edge information, and therefore, the results and edge reconstruction quality can be easily interpreted. The image is chosen to be of size $N \times N$ pixels, with $N = 200$, cf. Figure 1a. Since, according to the sampling condition (5), full aliasing free angular sampling requires $\lceil \pi N_s \rceil = 472$ samples in the $s$-variable, we computed tomographic data at $2N_s + 1 = 301$ equally spaced signed distances $s_\ell \in [-1.5, 1.5]$ and at $N_\varphi = 40$ equally spaced directions in $[0, \pi)$. This data is properly sampled in the $s$-variable, but undersampled in the angular variable $\varphi$, cf. Figure 1b. In all following numerical simulations, the regularization parameter $\lambda > 0$ and the tuning parameter $\mu \geq 0$ of (21) have been chosen manually. The development of automated parameter selection is beyond the scope of this paper.

From this data, we computed the approximate Laplacian reconstruction, shown in Figure 1c, using the standard FBP algorithm in combination with the LoG-filtered data $u_{\mathrm{LoG}} \circledast_s y_\Theta$ that we computed in a preprocessing step using the LoG data filter from Proposition 1. It can be clearly observed that FBP introduces prominent undersampling artefacts (streaks), so that many edges in the calculated feature map are not related to the actual image features. This shows that the edge maps computed by FBP (from undersampled data) can include unreliable information and even falsify the true edge information (since artefacts and actual edges superimpose). In a more realistic setup, this could be even worse, since artefacts may not be that clearly distinguishable from actual edges.

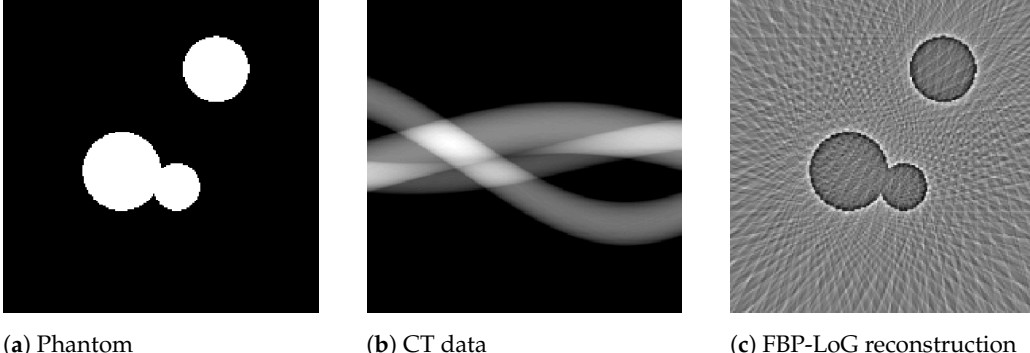

(**a**) Phantom            (**b**) CT data            (**c**) FBP-LoG reconstruction

**Figure 1.** RECONSTRUCTION OF THE LAPLACIAN FEATURE MAP USING FBP. The phantom image of size $200 \times 200$ consisting of a union of three discs (**a**) and the corresponding angularly undersampled CT data, measured at 40 equispaced angles in $[0, \pi)$ and properly sampled in the $s$-variable with 301 equispaced samples $s_\ell \in [-1.5, 1.5]$ (**b**). Subfigure (**c**) shows the Laplacian of Gaussian (LoG) reconstruction using the standard FBP algorithm. It can be clearly observed that FBP introduces prominent streaking artefacts that are due to the angular undersampling.

Figure 2 shows reconstructions of feature maps from noise-free CT data that we computed using our framework (21) for three different choices of feature extraction filters and for two different sets of regularization parameters. The first row of Figure 2 shows reconstructions with $\mu = 0$ and $\lambda = 0.001$ using 1000 iterations of the FISTA algorithm, whereas the second row shows reconstructions that were computed using an additional $H^1$-term with $\lambda = \mu = 0.001$ and using 500 iterations of the FISTA algorithm. In contrast to the FBP-LoG reconstruction (shown in Figure 1c), the undersampled artefacts have been removed in all cases. As expected, the use of $\ell^1$ regularization without an additional $H^1$ smoothing (shown in first row) produces sparser feature maps as opposed to the reconstruction shown in the second row. However, we also observed that the iterative reconstruction based only on the $\ell^1$ minimization (without the $H^1$ term) sometimes has trouble reconstructing the object boundaries properly. In fact, we found that a proper reconstruction of boundaries is quite sensitive to the choice of the $\ell^1$ regularization parameter. If this parameter was chosen to be too large, we observed that the boundaries could be incomplete or even disappear. Since the $\ell^1$ regularization parameter controls the sparsity of the reconstructed feature map, this observation is actually not surprising. By including an additional $H^1$ regularization term, the reconstruction results become less sensitive to the choice of regularization parameters.

In order to simulate real world measurements more realistically, we added Gaussian noise to the CT data that we used in the previous experiment. Using this noisy data, we calculated reconstructions via (21) in combination with the Ram–Lak-type filter (20) using three different sets of regularization parameters and 1000 iterations of the FISTA algorithm in each case. The reconstruction using the parameters $\lambda = 0$ and $\mu = 0.001$ (i.e., only $H^1$ regularization was applied) is shown in Figure 3a. The reconstruction in Figure 3b uses only $\ell^1$ regularization, i.e., $\mu = 0$ and $\lambda = 0.001$, and the reconstruction in Figure 3b applies both regularization terms with $\lambda = \mu = 0.001$. In both reconstructions shown in Figure 3b,c, the recovered features are much more apparent than for pure $H^1$ regularization. As in the noise-free situation, we observe that the (pure) $\ell^1$ regularization might generate discontinuous boundaries, whereas the combined $H^1$-$\ell^1$ regularization results in smoother and (seemingly) better represented edges. Note that a form of salt-and-pepper noise is observed in the reconstructions that include the $\ell^1$ penalty. We attribute this to the thresholding procedure within FISTA and the rather small regularization parameter. Increasing the regularization parameter would reduce the amount of noise, but would potentially remove some of the desired boundaries.

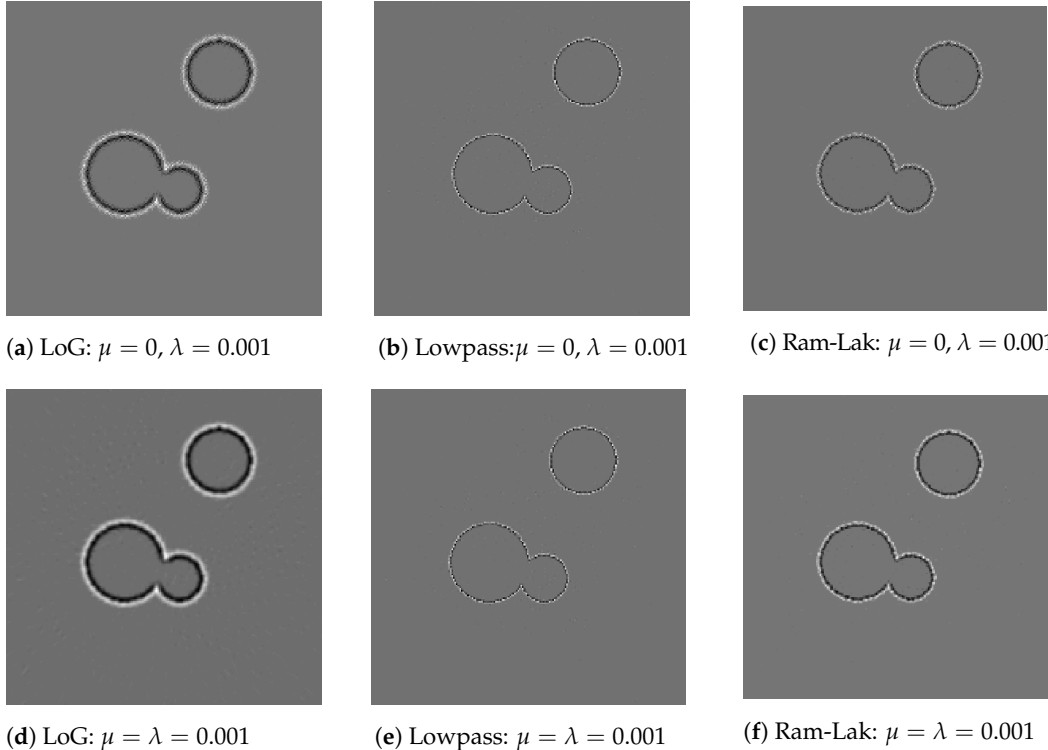

(**a**) LoG: $\mu = 0$, $\lambda = 0.001$     (**b**) Lowpass: $\mu = 0$, $\lambda = 0.001$     (**c**) Ram-Lak: $\mu = 0$, $\lambda = 0.001$

(**d**) LoG: $\mu = \lambda = 0.001$     (**e**) Lowpass: $\mu = \lambda = 0.001$     (**f**) Ram-Lak: $\mu = \lambda = 0.001$

**Figure 2.** RECONSTRUCTION OF LAPLACIAN FEATURE MAPS USING OUR FRAMEWORK. This figure shows reconstructions of feature maps from noise-free CT data that we computed using our framework (21) for three different choices of feature extraction filters and for two different sets of regularization parameters. Here, LoG refers to (12), low-pass to (18), and Ram–Lak to (20). The first row shows reconstructions with $\mu = 0$ and $\lambda = 0.001$ using 1000 iterations of the FISTA algorithm, whereas the second row shows reconstructions that were computed using an additional $H^1$ term with $\lambda = \mu = 0.001$ and using 500 iterations of the FISTA algorithm. In contrast to the FBP-LoG reconstruction (shown in Figure 1c), the undersampling artefacts have been removed in all cases.

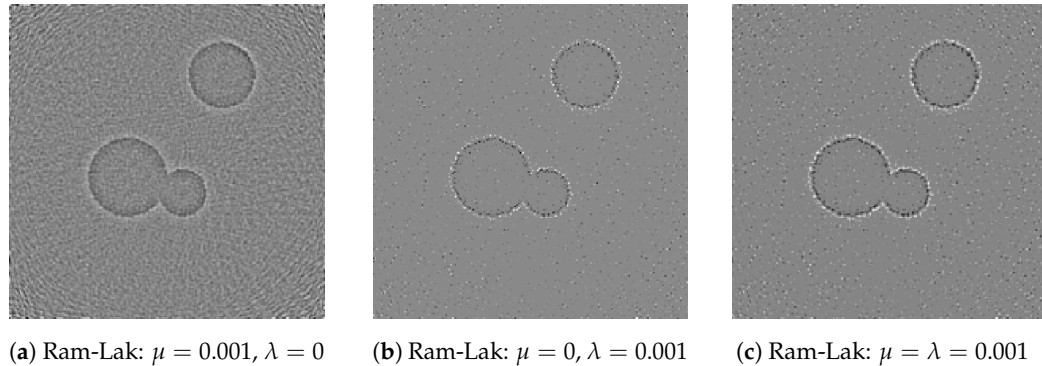

(**a**) Ram-Lak: $\mu = 0.001$, $\lambda = 0$     (**b**) Ram-Lak: $\mu = 0$, $\lambda = 0.001$     (**c**) Ram-Lak: $\mu = \lambda = 0.001$

**Figure 3.** RECONSTRUCTIONS OF LAPLACIAN FEATURE MAPS FROM NOISY DATA. The reconstruction in (**a**) was calculated using only $H^1$ regularization, in (**b**) using only $\ell^1$ regularization, and in (**c**) using combined $\ell^1$ and $H^1$ regularization.

### 4.2. Edge Detection

One main application of our framework for the reconstruction of approximate image gradients or approximate Laplacian feature maps is in edge detection. Clearly, feature maps that contain less artefacts can be expected to provide more accurate edge maps.

For this experiment, we used a modified phantom image that is shown in Figure 4a. In contrast to the previously used phantom, this image also includes weaker edges that are more challenging to detect. For this phantom, we generated CT data using the same

sampling scheme as in our first experiment (Section 4.1) and computed the LoG-feature maps $f \circledast U_{\mathrm{LoG}}$ using the FBP approach (cf. Figure 4b) and using our approach (cf. Figure 4c) with $\mu = 0$, $\lambda = 0.002$, and 100 iterations of the FISTA algorithm for (21). Subsequently, we generated corresponding binary edge maps by extracting the zero-crossings of these LoG-feature maps (cf. Figure 4d,e) by using MATLAB edge functions. Note that this procedure is a standard edge detection algorithm known as the LoG edge detector, cf. [30]. For both methods, we took a standard deviation of $\alpha = 1.3$ for the application of the Gaussian smoothing and a threshold of $t = 0.005$ for the detection of the zero crossings. As can be clearly seen from the results, the edge detection based on our approach (cf. Figure 4d) is able to also detect the weaker edges inside the large disc. In contrast, edge detection in combination with the FBP-LoG feature map was not able to detect the edge set correctly due to strong undersampling artefacts.

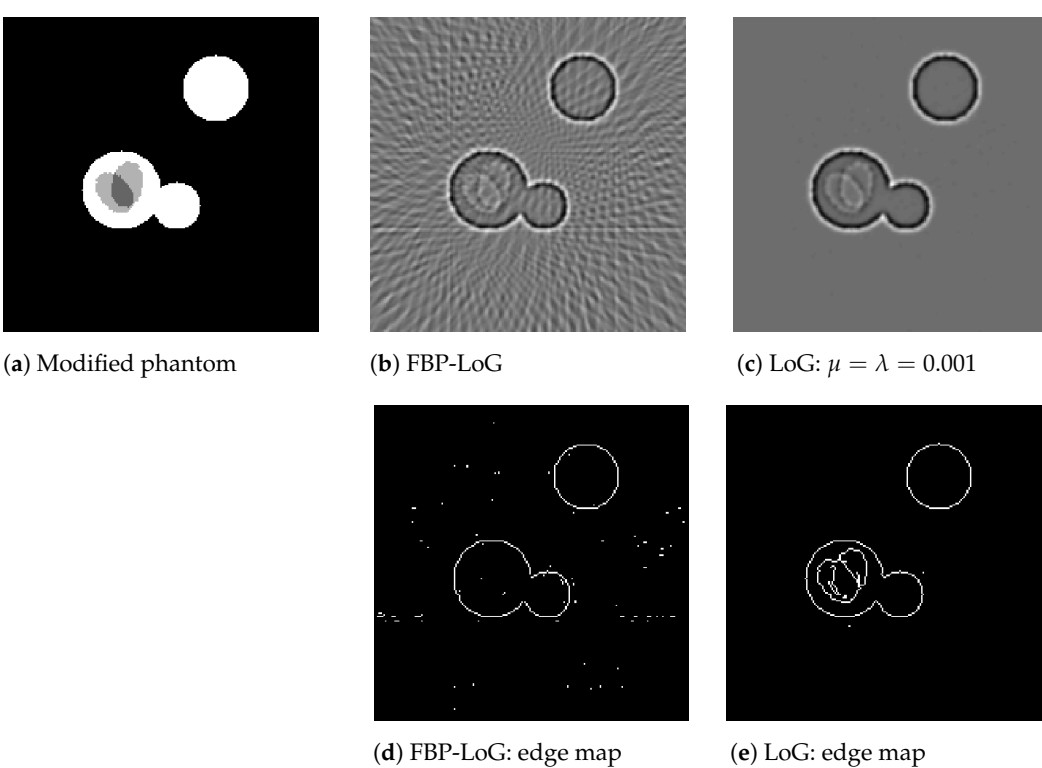

(**a**) Modified phantom      (**b**) FBP-LoG      (**c**) LoG: $\mu = \lambda = 0.001$

(**d**) FBP-LoG: edge map      (**e**) LoG: edge map

**Figure 4.** LoG EDGE DETECTION. The modified phantom image (**a**) also includes weaker edges that are more challenging to detect. Subfigures (**b**,**c**) show reconstructions of the LoG feature maps that were generated using the FBP algorithm and our approach, respectively. The corresponding binary edge masks generated by the LoG edge detector are shown in (**d**,**e**).

In our last experiment, we presented edge detection results for real, noisy CT scans of a lotus root [32]. This gives an estimation on the feature reconstruction quality for real life applications, i.e., much more complex data, where the sought feature maps are generally much more complicated compared to our synthetic phantom above. Note that similar reconstructions were presented in [12]. In order to obtain parallel-beam CT data that fit our implementation of $\mathbf{R}_\Theta$, we rebinned the lotus data (originally measured in a fan beam geometry) and downsampled it to $2N_s + 1 = 739$ signed distances and $N_\varphi = 36$ directions, cf. Figure 5d. The Gaussian gradient feature map was computed in two ways: firstly, by applying FBP to the filtered CT data with the data filter (11), cf. Figure 5b; and secondly, by using our approach (8) with $\mu = 0$ and $\lambda = 0.01$ and by applying 50 iterations of the FISTA algorithm, cf. Figure 5c. The resulting image size was $521 \times 521$. The standard deviation for the Gaussian smoothing was chosen as $\alpha = 6$, and for the Canny edge detection we used the same lower threshold 0.1 and upper threshold 0.15. In order to calculate binary edge maps (shown in Figure 5e,f), we used the Canny edge detector

(cf. [29]) in combination with the pointwise magnitude of the Gaussian gradient maps $|\nabla U_{\mathrm{grad}} \circledast f|$. Again, it was observed that the calculation of the Gaussian gradient map using our approach leads to more reliable edge detection results.

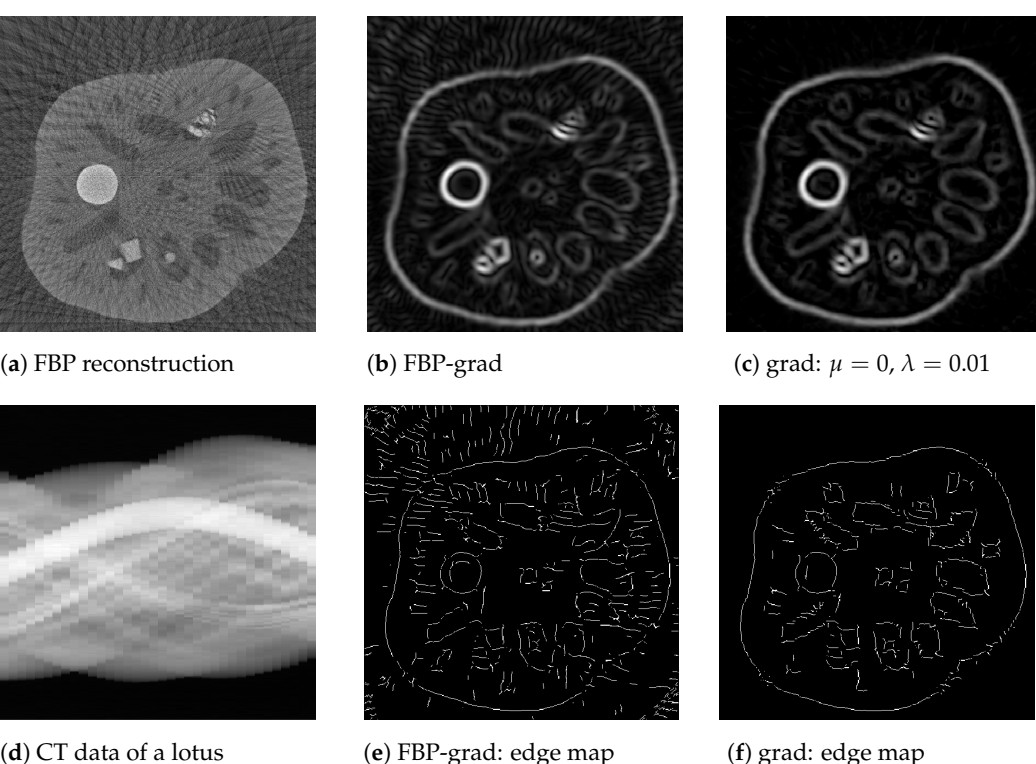

(**a**) FBP reconstruction      (**b**) FBP-grad      (**c**) grad: $\mu = 0$, $\lambda = 0.01$

(**d**) CT data of a lotus      (**e**) FBP-grad: edge map      (**f**) grad: edge map

**Figure 5.** Canny edge detection from the lotus data set. Rebinned CT data of a lotus root (**d**) (cf. [32]) and the corresponding FBP reconstruction (**a**) from 36 evenly distributed angles in $[0, \pi)$. Magnitude of the smooth gradient map $|\nabla U_{\mathrm{grad}} \circledast f|$ computed using the FBP algorithm (**b**) and using our approach (**c**). The corresponding edge detection results using the Canny algorithm are shown in (**e**) and (**f**), respectively.

**Remark 4.** *In all of our experiments, especially in Figures 1–4, we used phantoms that are piecewise constant images. Our intention here was to examine the performance of our method on phantoms with well-defined geometric edges. However, we would like to note that for such piecewise constant imagesl a two-step approach that combines total variation (TV) reconstruction and edge detection, is expected to produce excellent results, too. This is mainly because piecewise constant images are well represented by the TV-model.*

*In general, the performance of edge detectors that are realized within a two-step approach heavily relies on the a priori assumptions and on the use of suitable priors for the underlying image class. In contrast, our approach aims at reconstructing image features directly from CT data. Therefore, we only need to incorporate an a priori assumption about image features into our framework, which can be formulated independently of the underlying image class. In this sense, our approach is conceptually different from the two-step approach and can be applied in a general imaging situation. In case of edge-reconstruction form CT data, we have shown that a suitable a priori assumption is the sparsity of edge maps (in the pixel domain) and that these apriori assumptions can be efficiently incorporated into our framework by using the $\ell^1$-prior, yielding numerically efficient algorithms.*

## 5. Conclusions

In this paper, we proposed a framework for the reconstruction of features maps directly from incomplete tomographic data without the need of reconstructing the tomographic image $f$ first. Here, a feature map refers to the convolution $U \circledast f$ where $U$ is a given convolution kernel and $f$ is the underlying object. Starting from the forward convolution

identity for the Radon transform, we introduced a variational model for feature reconstruction, which was formulated using the discrepancy term $\|\mathbf{R}_\Theta h - u_\Theta \circledast_s y_\Theta\|_2^2$ and a general regularizer $r(h)$. In contrast to existing approaches, such as [9,10], our framework does not require full data and, due to the variational formulation, also offers a flexible way for integrating a priori information about the feature map into the reconstruction. In several numerical experiments, we have illustrated that our method can outperform classical feature reconstruction schemes, especially if the CT data is incomplete. Although we mostly focused on the reconstruction of feature maps that are used for edge detection purposes, our framework can be adapted for a wide range of problems. Specifically, such extensions of our framework require the convolutional features to satisfy certain equations that are derived from the data of the original inverse problem. Recently, such equations have been derived for photoacoustic tomography [33]. A rigorous convergence analysis of the presented scheme remains an open issue. Another direction of further research may include the extension of the proposed approach to non-sparse, non-convolutional features and generalization to other types of tomographic problems such as photoacoustic imaging [34]. Additionally, multiple feature reconstruction (similar to the method [33,35]) seems to be an interesting future research direction.

**Author Contributions:** S.G. carried out the numerical implementation and validation of the proposed approach. He also drafted the manuscript. M.H. and J.F. participated in designing and writing the article. All authors read and approved the final manuscript.

**Funding:** The work of M.H. was supported by the Austrian Science Fund (FWF) project P 30747-N32. The contribution by S.G is part of a project that has received funding from the European Union's Horizon 2020 research and innovation programme under the Marie Skłodowska-Curie grant agreement No 847476. The views and opinions expressed herein do not necessarily reflect those of the European Commission.

**Conflicts of Interest:** The authors declare no conflict of interest.

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
