# Peer review of "Feature Reconstruction from Incomplete Tomographic Data without Detour"

_mathematics, doi:10.3390/math10081318_

Round 1

Reviewer 1 Report

The developed method is of interest in the area and can be applied in the study of CT images.
The paper is very well organized and written.
The descriptions of the implemented methods are made very clearly and deductively.
The developed method is of interest in the area and can be applied in the study of CT images.
Some suggestions for improvement:

  • In the outline include a short sentence about Section 1.
  • Some words are poorly hyphenated, as in line 136, 251, 283, 285, 310, 347, 411.
  • In line 154, the equation is missing a closing parenthesis before the phrase "For the Randon transform we will ...."
  • In proposition 1, in item 2 in the last equation before line 286 replace "bzw." with "and".
  • On line 292 in the second equation add alpha to the function f (delta_x f_alpha).
  • In line 310 point 1 of Remark 3.4 the title is misspelled: replace "LOWPASS LALPACIAN" with "LOWPASS LAPLACIAN"
  • Add final point at the end of equation 3.11
  • The title of point 4.1 is misspelled. Fix the word map.

Author Response

Dear Reviewer 1,

thank you for your reviewing our manuscript and for your valuable comments. Please find our comments in the attached pdf-file.

Sincerely,

Jürgen Frikel

Reviewer 2 Report

This paper presents feature reconstruction from incomplete tomographic data.

The paper is an interesting approach in the aspect of using regularization method in feature reconstruction associated with limited data.

Minor remarks:

1) I suggest describing in more detail on what basis and from what the scope of selection of input data for the research problem posed.

2) The authors could refer to other research methods used to reconstruct images from incomplete data and justify the choice of the method presented.

3) The article needs minor linguistic corrections.

Author Response

Dear Reviewer 2,

thank you for reviewing our manuscript and for your valuable comments. Please find our comments in the attached pdf-file.

Sincerely,

Jürgen Frikel

Reviewer 3 Report

The paper describes a novel framework for a single step CT feature reconstruction in contrast to the conventional two-step approach (CT volume reconstruction followed up by the edge detection). As the edge detection is typical post-processing routine applied to CT data, the method is a valuable contribution for many CT applications. Please, consider to address the comments and suggestions below.

General comments:

- The application domain of the developed method should be more clearly stated. Low dose medical imaging is indeed an important area of research. However medical images are typically characterized by low contrast and high-texture regions. The robustness of the method in this scenarios is not demonstrated.

- The advantage of the proposed 1-step method versus conventional 2-step approach (for instance conventional FISTA TV CT volume reconstruction followed up by edge detection) should be clearly articulated and demonstrated.

- The extension of the method to other scanning geometries (cone/fan beam) should be discussed.

Other comments and suggestions:

Consider checking affiliations: 1 and 3 are identical, the first's author email is duplicated in 1 and 3

line7: firs -> first

abstract: the abstract would benefit from a sentence or two on 'non-linear (variational) regularization methods' for non-familiar readers

introduction: low-dose scanning scenario (low SNR/CNR and angular undersampling) is relevant not only to medical CT but also to in-vivo scanning for biological applications and for fast tomography in general

general comment: X in X-rays is generally capitalized

l37: "we consider the parallel beam geometry" as far as I know the parallel beam case is more relevant to synchrotron, i.e., non-medical CT; in medical CT cone or fan beam are typically employed. Please clarify why parallel beam geometry is considered here.

l43 "or a small number of directions" please clarify that you consider circular scanning trajectory

l52 "with uniformly distributed directions θ1, . . . , θm on a half-sphere" now I am getting confused: I've never seen such a scanning scenario in medical CT. If the scope of the paper is the theoretical development of the method rather it's practical applicability, please, state this clearly in the abstract and in the introduction.

360 " By including an additional H1-regularization term, the reconstruction results become less sensitive to the choice of regularization parameters." ... but still are there any procedures to find the regularization parameter?

figure 4.2. How these results would compare to a more conventional FISTA TV reconstruction followed up by edge detection? By the conventional FISTA TV I mean CT reconstruction based on discrepancy between the acquired and forward projected data with TV regularization?

362 "we added noise to the CT data" Which kind of noise?

figure 4.3 I can observe quite significant 'salt and pepper' type of noise in the reconstructed images. Please comment on this. Secondly, I can imagine that with conventional FISTA TV reconstruction this noise heavily suppressed especially for such a simple texture-free binary phantom.

section 4.2 again, a comparison with conventional FISTA TV would make this section more complete.

393 As far as I know the lotus dataset has cone beam geometry. In the paper you are discussing only parallel beam geometry. Which adaptations of the method are required for the cone-beam case?

419 "framework can be adapted for a wide range of problems" Please provide examples

421 "may may" remove repetition

422 " other types of tomography problems" Please provide examples

Author Response

Dear Reviewer 3,

thank you for reviewing our manuscript and for your valuable comments. Please find our comments in the attached pdf-file.

Sincerely,

Jürgen Frikel

Round 2

Reviewer 3 Report

I would like o thank authors for addressing my comments and suggestions. The paper can be published. now.